

# Stratospheric residence time and the lifetime of volcanic stratospheric aerosols

Matthew Toohey[1], Yue Jia[2,3,1], Sujan Khanal[1], Susann Tegtmeier[1]

[1] Institute of Space and Atmospheric Studies, University of Saskatchewan, Saskatoon, S7N 5A2, Canada
[2] Cooperative Institute for Research in Environmental Sciences (CIRES), University of Boulder, USA
[3] NOAA Chemistry Sciences Laboratory, USA

*Correspondence to*: Matthew Toohey (matthew.toohey@usask.ca)

**Abstract.** The amount of time that volcanic aerosols spend in the stratosphere is one of the primary factors influencing the climate impact of volcanic eruptions. Stratospheric aerosol persistence has been described in different ways, with many works quoting an approximately 12 month "residence time" for aerosol from large tropical eruptions. Here, we aim to develop a framework for describing the evolution of global stratospheric aerosol after major volcanic eruptions and quantifying its persistence, based on global satellite-based aerosol observations, tracer transport simulations and simple conceptual modeling. We show that the stratospheric residence time of air, which is estimated through passive tracer pulse experiments and is one factor influencing the lifetime of stratospheric aerosols, is strongly dependent on the injection latitude and height, with an especially strong sensitivity to injection height in the first four kilometers above the tropical tropopause. Simulated stratospheric tracer evolution is best described by a simple model which includes a lag between the injection and initiation of removal from the stratosphere. Based on analysis of global stratospheric aerosol observations, we show that the stratospheric lifetime of stratospheric aerosol from the 1991 Pinatubo eruption is approximately 22 months. We estimate the potential impact of observational uncertainties on this lifetime finding it unlikely the lifetime of Pinatubo aerosol is less than 18 months.

## 1 Introduction

Volcanic eruptions that inject large amounts of sulfur-bearing gases into the atmosphere are a dominant natural driver of climate variability. Sulfur gases emitted from eruptions are converted to sulfate aerosols, which scatter incoming solar radiation and absorb infrared radiation, with the net result of a decrease in radiative flux at the surface and cooling (Robock, 2000). The cumulative climate impact of an eruption is related to the amount of sulfur emitted, but also to the amount of time the resulting aerosol particles spend in the atmosphere before being deposited to the Earth's surface. While the lifetime of sulfate aerosols in the troposphere is on the order of days to weeks, it is much longer in the stratosphere, due to the lack of wet deposition and relatively slow mixing with the troposphere (Boucher, 2015).





Stratospheric aerosol enhancements can persist for several years, but quantification of this persistence can differ. The IPCC AR5 describes the lifetime of stratospheric aerosols to be around 1 year for tropical eruptions, and 6-9 months[1] for high-latitude eruptions (Myhre et al., 2013). Another IPCC AR5 chapter states that for tropical eruptions, the aerosol cloud "lasts between one and two years" (Kirtman et al., 2013). Robock (2000) and Thomason and Peter (2006) describes an "e-folding residence time of about 1 year" for aerosols from large tropical volcanic eruptions. The observational studies which these estimates are

based on are often more specific in their quantification of aerosol persistence. For example, McCormick et al. (1995) state that "since the middle of 1992, the total stratospheric aerosol mass has decreased with a 1/e-folding time of approximately 1 year". Similarly, based on lidar measurements of stratospheric aerosol over Mauna Loa, Barnes and Hofmann (1997) report that aerosol levels decreased with a "characteristic exponential decay time" of 1 year after both the 1991 Pinatubo and 1982 El Chichón eruptions, measured over a period 6 months to 3.5 years after the eruptions. Note that in both cases, these studies

report a timescale describing the rate of decay of aerosol (rather than a "lifetime" or "residence time") over a period that begins 6-12 months after the eruption.

The spreading of volcanic sulfur and sulfate aerosols through the stratosphere and removal from the stratosphere are understood to be controlled primarily by two physical processes. First, aerosol particles are moved by stratospheric winds and the large-scale circulation, named the Brewer-Dobson circulation (BDC), which is characterized by upwelling in the tropics, mixing and

poleward transport in the mid-latitudes, and downwelling over the high latitudes (Butchart, 2014; Shepherd, 2007). Observations of stratospheric aerosol have been used to confirm the relative confinement of air in the so-called "tropical pipe" (Plumb, 1996; Trepte and Hitchman, 1992), where air is only slowly mixed into the extratropical regions of both hemispheres (Trepte et al., 1993). The BDC is seasonally dependent, with stronger poleward motion and downwelling in the winter hemisphere (Rosenlof, 1995). Removal from the stratosphere is believed to occur primarily via pseudo-horizontal mixing

across the midlatitude tropopause (Hamill et al., 1994). Secondly, aerosols can move vertically relative to the air around them due to gravitational settling. Theoretical treatment implies that vertical fall velocity varies strongly with both aerosol size and altitude (Junge et al., 1961). Based on aerosol observations from balloon-borne optical particle counter instruments, Hofmann and Rosen (1983) presented evidence to support the theory of a size segregation of aerosols, with larger aerosols being removed more rapidly from the stratosphere than smaller aerosol after the El Chichón eruption.

The processes controlling the persistence of stratospheric aerosol and the most useful ways of quantifying it are poorly understood. While observations show a difference in the global transport and persistence of stratospheric aerosol from the large tropical eruptions of Pinatubo and El Chichón compared to high-latitude eruptions, it remains unclear to what degree this is due to the latitude of the injection or the injection height (Toohey et al., 2019). The relative importance of stratospheric circulation vs. gravitational settling on aerosol persistence remains unquantified. Aerosol-climate models produce wide

---

[1] A month is a rather imprecise unit of measurement, since calendar months contain different numbers of days. We use the unit here, however, since it has been used extensively in prior relevant studies, it is rather intuitive for the timescales of interest here, and since uncertainties in observations limit the precision of results anyhow. For the record, we define one month as equivalent to 365/12=30.4 days.





variation in the temporal evolution of stratospheric aerosol mass burden (Clyne et al., 2021; Marshall et al., 2018; Quaglia et al., 2023), and we lack a framework to quantify the differences in aerosol persistence. These gaps have implications for our ability to gain understanding from aerosol observations of recent eruptions as well as model simulations of volcanic eruptions and geoengineering scenarios (Visioni et al., 2023; Tilmes et al., 2017; Sun et al., 2023).

In this work, we aim to develop a framework to understand the lifetime of stratospheric aerosol and provide a robust estimate

of the lifetime of aerosol from the 1991 Mt. Pinatubo eruption. We also explore the dependency of stratospheric residence time on the injection latitude, height and season. The remainder of this manuscript is organized as follows. In Sect. 2, we introduce some theory on the quantification and measurement of residence time and extend these concepts to consideration of stratospheric aerosol. In Sect. 3, we introduce the stratospheric aerosol data and model simulations analyzed in this work. Sec 4 includes a presentation of the results. Conclusions and discussion follow in Sect. 5.

## 2 Theory

### 2.1 Residence time

The concept of residence time has wide applicability to natural science, and has a long history in the field of chemical engineering in the specific context of quantifying the residence time of fluids in tanks or reactors (Fogler, 2020), but also in studies of geophysical systems like lakes (e.g., Ambrosetti et al., 2003) or the stratosphere (Hall and Waugh, 2000).

Consider a reservoir in which there is flow from input to output. Every fluid element that enters the reservoir will spend some time within the reservoir before eventually exiting it: the time spent within the reservoir defines the residence time of each fluid element. The frequency of occurrence of the residence time $t$ in the set of all the particles that are leaving the reservoir is quantified by the residence time distribution $E(t)$. All fluid must eventually leave the reservoir, therefore $\int_0^\infty E(t)dt = 1$.

The mean residence time of the particles leaving the reservoir is the average of all the residence times of the particles leaving

the reservoir. It can be calculated as the first moment of the residence time distribution, i.e.:

$$\tau_r = \int_0^\infty tE(t)dt. \tag{1}$$

The term "residence time" is, we suggest, often used as short-hand for "mean residence time" as defined above.

If a reservoir has a single input, a single residence time distribution (RTD) may be satisfactory to characterize the reservoir. If a reservoir has multiple inputs, a mean residence time may be defined for each input separately. In many geophysical contexts, it may be useful to define a mean residence as a continuous function of location within the reservoir. Here we treat mean

stratospheric residence time as a function of the injection (or emission) location (Hall and Waugh, 2000), applicable specifically to injection of material from major volcanic eruptions.

The residence time distribution or other related metrics can be determined experimentally for a reservoir. One common method is the "pulse input experiment", in which an amount of tracer is suddenly injected into the reservoir. In chemical reactor studies,





it is common to measure the concentration of the injected tracer at the reactor output, from which the residence time distribution

is easily determined. In some cases, we may monitor the fraction of the tracer remaining in the reservoir as a function of time, which we call the washout function $W$. At any time, the washout function is simply the total fraction injected (1) minus the amount of tracer which has left the reservoir:

$$W(t) = 1 - \int_0^t E(s)ds. \qquad (2)$$

Therefore, if the washout function is observed, the residence time distribution is

$$E(t) = -\frac{dW}{dt}. \qquad (3)$$

Residence time distributions can be defined analytically for certain idealized scenarios. In "plug flow", fluid flow is modeled as a series of "plugs" traveling along the axial direction of a cylinder. When such a system would be used in a pulse experiment, the fraction of injected tracer in the system would remain at unity until the plug of tracer exits the reservoir all at once (Fig 1a). The residence time distribution of a plug-flow reservoir is therefore a Dirac delta function centered on the mean residence time (Fig 1d).

If a reservoir is well-mixed, such that the concentration of any tracer is instantaneously homogeneous within the reservoir upon injection, then the rate of tracer removal from the reservoir is proportional to the amount of tracer in the reservoir. This defines a "first order process", for which the mathematical representation of tracer amount as a function of time is exponential decay, i.e.:

$$W(t) = \exp(-t/\tau_d), \qquad (4)$$

where the rate of decay is described by the decay time constant $\tau_d$. In a pulse experiment to an idealized well-mixed reservoir,

the concentration of tracer within the reservoir will decay exponentially with a decay time constant $\tau_d$ (Fig 1b). For this special and idealized case, the residence time distribution is also exponential in nature (Fig 1e), and the mean residence time is equal to the decay time constant $\tau_d$, which is equal to the time required for the amount of tracer to cross a value of 1/e of its initial amount, also known as the "e-folding time".

Finally, we introduce a third idealized case which will be relevant to the discussion of stratospheric aerosol below. Imagine

that when a tracer is introduced to the reservoir, the tracer is well-mixed within a finite volume which increases in size with time. After some time lag $\tau_l$, the well-mixed volume envelopes the outlet of the reservoir, upon which time exponential decay of the amount of tracer in the reservoir begins. In this case, the washout function $W$ takes the form of a constant value of 1 while the well-mixed volume expands toward the exit, then follows exponential decay thereafter (Fig 1c). The residence time distribution takes the form of an exponential decay, shifted in time by the time lag before decay begins (Fig 1f). In mathematical

form, the washout function for this lagged-decay model can be expressed as:



$$W = \begin{cases} 1, & t < \tau_l \\ \exp\left(-\dfrac{t - \tau_l}{\tau_d}\right), & t \geq \tau_l \end{cases} \tag{5}$$

Note that this expression simplifies to the well-mixed tank and plug flow described above with suitable choices of parameters (e.g., $\tau_l = 0$ for the well-mixed exponential decay model and $\tau_d = 0$ for plug flow). Taking the derivative and calculating the first moment as in Eq. 1, we find that the residence time of a tracer described by this lagged-decay model is $\tau_r = \tau_l + \tau_d$. For

example, in the idealized example shown in Fig. 1c, we have a 12-month lag until exponential decay begins with a decay timescale of 12 months. In this case, the mean residence time is $12 + 12 = 24$ months. Similar to the simple well-mixed scenario, the mean residence time is equal to the e-folding time, measured from the time of the tracer injection.

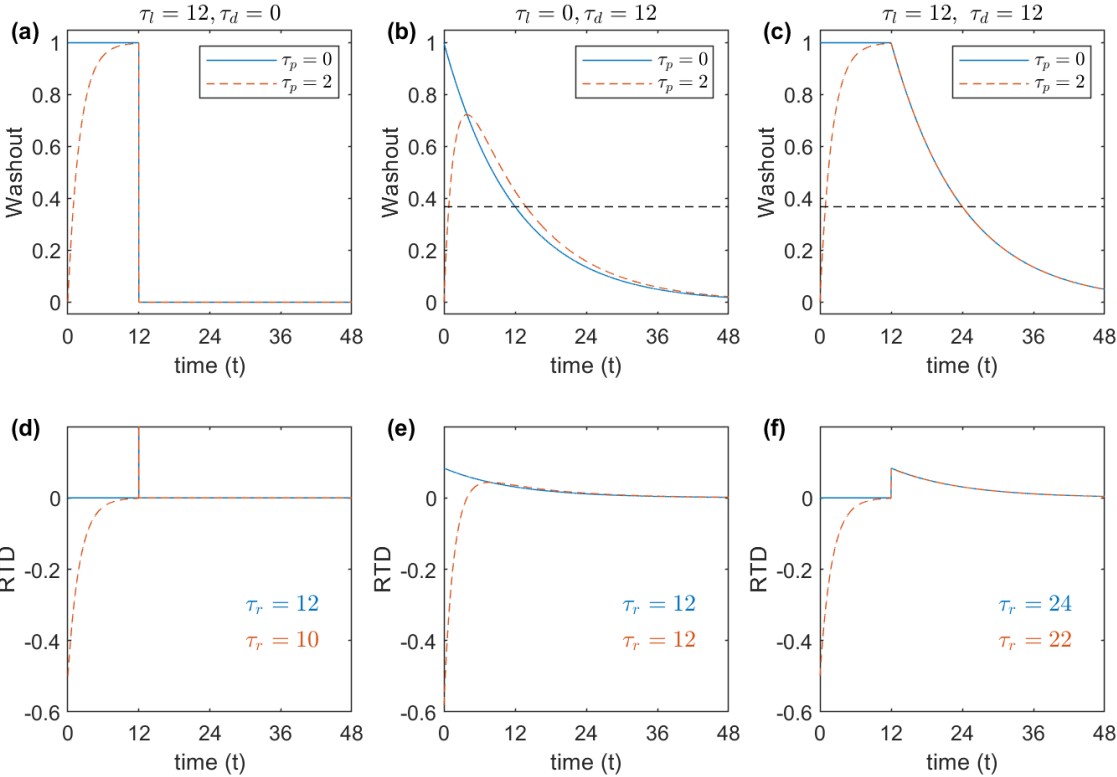

**Figure 1: Washout functions and associated residence time distributions (RTDs) for idealized pulse experiments for three idealized reservoir flow scenarios: plug flow (a,d), well-mixed (b,e) and lagged decay (c,f). Blue lines represent the washout function and RTD of a primary tracer injected into the reservoir, red lines represent quantities for a secondary tracer produced in the reservoir from the primary tracer with timescale $\tau_p$. All washout functions are described by Eq. 6 with parameter values indicated by title text and legends to panels a, b and c. For each scenario, the residence time $\tau_r$ is given in bottom panels.**



## 2.2 The lifetime of a secondary tracer

The climate impact of volcanic eruptions is primarily the result of the injection of sulfur to the stratosphere, which is initially mostly in the form of $SO_2$. $SO_2$ is oxidized to $H_2SO_4$ over a timescale of days-to-weeks, which condenses to liquid sulfate aerosol particles. The particles can grow in size, through condensation of gaseous $H_2SO_4$, and through coagulation. The aerosol particles are advected by stratospheric winds, and as they grow larger, may experience appreciable vertical motion as a result of gravitational settling. Eventually, the particles exit the stratosphere across the tropopause. Once in the troposphere, it is assumed that they are quickly scavenged and deposited to the surface.

The injection of volcanic material into the stratosphere is in some ways similar to the pulse injection experiment used to measure the residence time distribution of a chemical reactor. On the other hand, the volcanic pulse experiment differs from laboratory experiments in a few key ways. Firstly, since aerosol particles may gravitationally settle, they may persist in the stratosphere for less time than does the air in which they originally formed. Thus, we differentiate between the residence time of air, which would be measured by a passive tracer in a pulse experiment, and the residence time of volcanic sulfur. The two differ because of the gravitational settling of sulfur-containing aerosol particles. This means that the residence time of stratospheric sulfur is likely to be not only a function of injection location but also of the magnitude of sulfur injection, since larger injections may lead to larger aerosol sizes and faster fallout.

Secondly, since aerosols are formed sometime after the injection of sulfur to the stratosphere, we make the distinction between the residence time of stratospheric sulfur, and the lifetime of stratospheric aerosols. (The terms "residence time" and "lifetime" are often used interchangeably, lifetime being preferred when the loss process is chemical rather than due to flow out of a reservoir (Jacob, 1999). We use lifetime here to reinforce the point that stratospheric aerosol is produced via chemical and microphysical processes after the injection of the precursor gases.)

If one were able to track the amount of total sulfur in the stratosphere as a function of time, it may look similar to the idealized pulse experiment washout functions shown in blue lines in Fig. 1(a,b,c). The amount of secondary tracer, on the other hand, will start at 0 and increase with time according to the production timescale $\tau_p$, as in the red lines in the examples shown in Fig. 1(a,b,c). The washout function for the secondary tracer is the ratio of the amount of secondary tracer at any time to the total amount of secondary tracer which is removed from the reservoir, not the amount of primary tracer injected—the two may differ if some of the primary tracer is removed before conversion. We derive (see Appendix 1) the washout function for a secondary tracer, which we refer as a production-lag-decay (PLD) model:

$$W_2 = \begin{cases} \dfrac{1}{F_2}\left[1 - \exp\left(-\dfrac{t}{\tau_p}\right)\right], & t < \tau_l \\[2ex] \dfrac{1}{F_2}\left[1 - \exp\left(-\dfrac{t}{\tau_p}\right)\right]\exp\left(-\dfrac{t - \tau_l}{\tau_d}\right), & t \geq \tau_l \end{cases} \tag{6}$$

where $F_2$ represents the fraction of the initial tracer injection removed from the reservoir as the secondary tracer:



$$F_2 = 1 - \frac{\tau_p}{\tau_d + \tau_p} \exp\left(-\frac{\tau_l}{\tau_p}\right). \tag{7}$$

Note that as $\tau_p \to 0$, $F_2 \to 1$ and this expression for the washout function approaches the simpler expression in Eq. 5.

Differentiating $W_2$ and taking the first moment to compute the lifetime of the secondary tracer results in:

$$\tau_r = \frac{\tau_l + \tau_d}{F_2} - \tau_p \tag{8}$$

This expression for the lifetime simplifies for certain cases discussed above. For example, with $\tau_l = 0$, we get $\tau_r = \tau_d$. This represents the well-mixed case, wherein the tracer is assumed to be instantaneously mixed within the reservoir and so the timescale of production does not affect the lifetime since it does not matter where in the reservoir the conversion of the primary to secondary tracer occurs. Also, if the lag timescale is longer than the production timescale $\tau_l \gg \tau_p$, then by Eq. 7, $F_2 = 1$ and by Eq. 8, $\tau_r = \tau_d + \tau_l - \tau_p$. This represents the case where all primary tracer is converted to secondary tracer before the
removal of tracer from the reservoir begins.

## 3 Methods

### 3.1 Tracer pulse experiment simulations

To explore stratospheric residence time, we used the FLEXible PARTicle (FLEXPART) dispersion model (Stohl et al., 2005), which is an offline model driven by 3-D meteorological fields. FLEXPART computes trajectories of a large number of particles
to represent the transport of mean flow as well as diffusive transport. In this study, we use FLEXPART version 10.0, which is driven by 6-hourly meteorological fields from ECMWF (European Centre for Medium-Range Weather Forecasts) reanalysis product ERA-Interim (Dee et al., 2011) with a 1° x 1° horizontal resolution and 61 vertical model levels.

In each pulse experiment, 100,000 passive tracers are initialized from a given latitude, longitude and height. The tracers are then advected forward in time for approximately 5 years. The tracers are passive, meaning neither loss processes (chemical
decay or deposition) nor gravitational settling of the particle is considered. Daily outputs of trajectories for each particle as well as corresponding atmospheric parameters (e.g., temperature, pressure, tropopause height) are recorded. A stratospheric residence time for each tracer is found by determining the first time at which the tracer's height is found to be below the local tropopause height, determined from the meteorological reanalysis data.

Pulse experiments are performed with injections at varying stratospheric heights and locations, spanning 0-60°N in steps of 5°
and 13 to 26 km in steps of 2 km. This residence time mapping exercise is performed for injections in boreal summer and winter, with start dates of June 15 and Dec 15, respectively. The meteorological input to FLEXPART is transient, and initialized from the year 1991, allowing the summer injections to be comparable to the observed evolution of sulfate from the Pinatubo eruption.





For each tracer simulation, an e-folding time is defined as the period required for the fraction of the initial tracer pulse
remaining in the stratosphere to cross 1/e=0.368. We also calculate a decay timescale as a function of time, based on a moving
11-month exponential fit to the stratospheric tracer fraction. A residence time distribution is calculated based on the time
derivative of the stratospheric fraction time series as in Eq. 3. Since the tail of the RTD to long residence times can have a
small but non-negligible impact on the mean residence time, we extend the RTD beyond the 5 years of the simulations by
assuming an exponential decay in stratospheric tracer fraction using a decay time constant of 20 months, which was consistent
with the decay timescale all simulations approached by the end of the 5-year simulations. A mean residence time for each
simulation was computed from the extended RTD using Eq. 1.

## 3.2 Observations

Volcanic $SO_2$ emissions are taken from the satellite base data set of Carn (2020). Derived from nadir-viewing satellite
instruments, the $SO_2$ estimates represent total columns, not limited to the stratospheric portion of injection. For the 1991
eruption of Pinatubo, Carn (2022) estimates a total emission of 15.0 $TgSO_2$, equivalent to 7.5 TgS. Uncertainty in the $SO_2$
emissions of Carn (2022) database are quoted as 20-30%. Taking the upper limit, we thus take as a best estimate the total $SO_2$
emission from Pinatubo to be $7.5 \pm 2.2$ TgS.

Based on inverse modeling of the Mt. Pinatubo cloud, Ukhov et al. (2023) estimate that 65% of the total sulfur emission from
Pinatubo was into the stratosphere, with the remaining amount injected below the tropopause where it was quickly removed.
Combining the result of Ukhov et al. (2023) with the total estimate of Carn (2022), we take $5.0 \pm 1.5$ TgS as the best estimate
of the stratospheric sulfur injection amount from the Pinatubo eruption. This value is consistent with some modelling studies
which have found best agreement with observations of the Pinatubo aerosol using a stratospheric injection of around 5 TgS
(Mills et al., 2016; Dhomse et al., 2014).

Aerosol extinction is taken from the global space-based stratospheric aerosol climatology GloSSAC v2.2, covering the years
1979-2021 (Kovilakam et al., 2020; Thomason et al., 2018). GloSSAC is constructed primarily from satellite-based
observations of the atmospheric limb. From 1979-2005, the primary data sources are the SAGE series of solar occultation
instruments. Observations of Pinatubo aerosol are based largely on measurements by the SAGE-II instrument. However, the
SAGE-II record contains gaps following the Pinatubo eruption to mid-1993 due to the extreme opacity of the stratosphere.
Such gaps in the SAGE record have been filled by complementary satellite data products as well as ground and airplane based
observations (Thomason et al., 2018).

We use GloSSAC v2.2 multispectral aerosol extinction data to derive an estimate of the stratospheric aerosol mass. Aerosol
mass content $m$ (units: $g \, m^{-3}$) can be related to measured aerosol extinction coefficient ($\beta$) (e.g., Schulte et al., 2023; Grainger,
2023):

$$m = \frac{4\rho\beta}{3 \, \tilde{Q}^{ext}} r_e,$$ (9)



which depends also on the aerosol density ($\rho$), effective radius ($r_e$) and the extinction efficiency ($\tilde{Q}^{ext}$), itself a function of the aerosol size distribution. The density of sulfuric acid solution ($\rho$), which for a given temperature varies with the concentration by weight, is calculated based on the parameterization described by Sandvik et al. (2019). If we assume a lognormal unimodal size distribution, $\tilde{Q}^{ext}$ is determined by $r_e$, so we can determine the mass density with an estimate of effective radius from the observations.

We determine the effective radius from the GloSSAC data following a method used by prior studies (Yue and Deepak, 1983). First, GloSSAC-based extinction ratio between the 525 and 1020 nm wavelength bands is calculated as a function of month, latitude and altitude. Second, using standard Mie code, $\tilde{Q}^{ext}$ for sulfuric acid droplets at the two given wavelengths is calculated as a function of $r_e$ assuming a lognormal distribution with a given distribution width ($\sigma$), sulfuric acid concentration by weight of 75%, and a characteristic stratospheric temperature ($T = 215$ K). Then the extinction efficiency ratio between

525 and 1020 nm as a function of effective radius is calculated. Since this relationship is a monotonic function of effective radius for values less than about 0.7 µm, it allows the estimation of effective radius with the extinction ratio from the GloSSAC data. The dependence on aerosol number concentration is removed since the ratio of the extinction coefficients at two wavelengths is considered.

With effective radius estimated as described above, we use GloSSAC extinction coefficient at 525 nm and the corresponding

extinction efficiency from the Mie code to estimate the aerosol mass content from Eq. 9. It is then integrated vertically to derive column aerosol mass density (units: $\mathrm{g\,m^{-2}}$), which is then converted to mass by multiplying by the area of each zonal bin. Finally, the mass is summed over the globe to produce a global aerosol mass. This process is repeated using a range of values for the distribution width ($\sigma = 1.2, 1.4, 1.6$) to represent uncertainty and variability in this parameter (Grainger et al., 1995).

Interpretation of aerosol evolution in the period after the June 1991 Mt. Pinatubo tropical (15°N) eruption is complicated by the high-latitude SH (46°S) eruption of Cerro Hudson in August of the same year. Analysis of the GloSSAC extinction in the months following the 2 eruptions suggests a minor contribution (<5%) of Cerro Hudson to the total stratospheric aerosol burden (see Appendix B). Accordingly, we treat the derived aerosol burden timeseries as dominated by the Pinatubo eruption in the following.

For comparison with our GloSSAC-derived mass timeseries, near-global (80°S - 80°N) sulfate aerosol mass estimates are also taken from Baran and Foot (1994), retrieved from measurements by the high-resolution infrared radiation sounder (HIRS).

## 4. Results

### 4.1 Passive tracer pulse experiments – residence time of air

The temporal spread of tracers in the zonal mean latitude-altitude plane is shown for an example FLEXPART pulse injection

experiment case in Fig. 2. One month after the tracer injection, the tracers are contained to the tropical pipe, spread between

 

~30°S to 30°N and from 18 to 28 km height. Three months after the eruption, the majority of tracers remain in the tropics, although a fraction have been transported to the SH, mostly at an altitude of around 20 km. By 6 months after the injection, tracers have spread to the SH pole, and into the NH midlatitude, although the tracer density remains highest in the tropics. Twelve months after injection, tracers are relatively well spread latitudinally, and peaking vertically around 20 km in the

extratropics and around 28 km in the tropics. As time passes to 24 and 48 months after the injections, the effect of the BDC circulation is apparent, pushing air masses from equator to pole and eventually downward into the lowermost stratosphere. At 48 months, the extratropical tracer density decreases exponentially with height with a scale height of 8.8 km, similar to the atmospheric scale height, indicating a well-mixed tracer of approximately uniform mixing ratio throughout the vertical extent of the atmosphere.

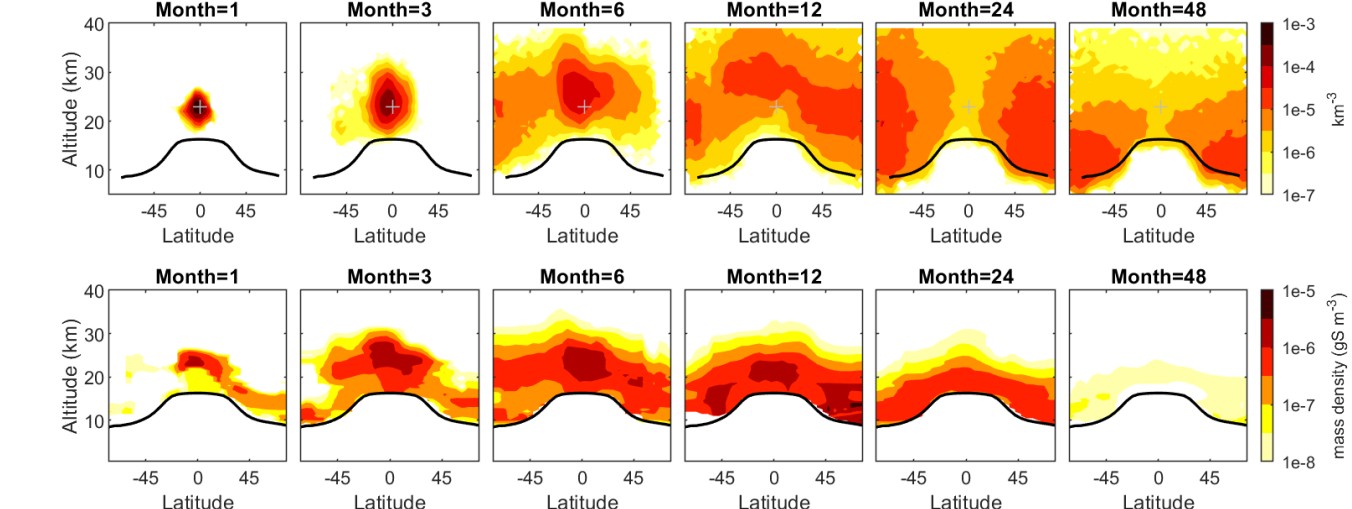

**Figure 2: Simulated tracer and observed aerosol spread in the meridional-vertical plane as a function of time. (Top) Zonal mean tracer density (km$^{-3}$) for months after injection as labeled for passive tracer pulse experiment with injection at 0°N and 23 km. The injection location indicated by grey cross in each panel. (Bottom) Aerosol mass density derived from GloSSAC after the 1991**
**Pinatubo eruption for months after eruption as labeled. The climatological zonal mean tropopause height is shown in each panel in black.**

Diagnostics of stratospheric tracer abundance from two example pulse experiments are shown in Fig. 3. The fraction of tracers remaining in the stratosphere represents a washout function as described in Sect. 2, and is shown in Fig. 3a for an injection at

20°N, 21 km. The washout function begins with a constant value of 1, as the trajectories spread through the stratosphere before reaching the tropopause. Appreciable cross-tropopause transport, and reduction of the washout function begins around month 7, and by month 12, the decay is roughly exponential in nature. The e-folding time is defined as the point at which $W$ crosses the 1/e threshold, and is found to be 26 months in this case. For an injection at 19 km and 40°N, the washout function (Fig 3b) shows a similar structure, albeit with a more rapid initiation of decay and a steeper decay, with an e-folding timescale of 17

months.



The time evolving decay timescale (Fig 3c,d) is calculated as the running 11-month exponential fit to the washout function. For the 21km, 20°N injection, by 12 months after injection, the decay timescale decreases to a value of approximately 20 months, and thereafter oscillates seasonally around this value. For the 19 km, 40°N injection case, the decay timescale minimizes at around 11 months after the eruption with a value of ~12 months, and thereafter increases and oscillates around a

value of 20 months from around 30 months after injection onward. We find that in all injection experiments, after 36-48 months the decay timescale tends to oscillate around the value of 20 months, suggesting that this is a representative timescale of stratospheric residence time for a well-mixed tracer. This result (equivalent to 608 days) is comparable to the 576 day aerosol decay timescale found by Sun et al. (2024) in their Lagrangian model experiment after a 10-year stratospheric aerosol injection period.

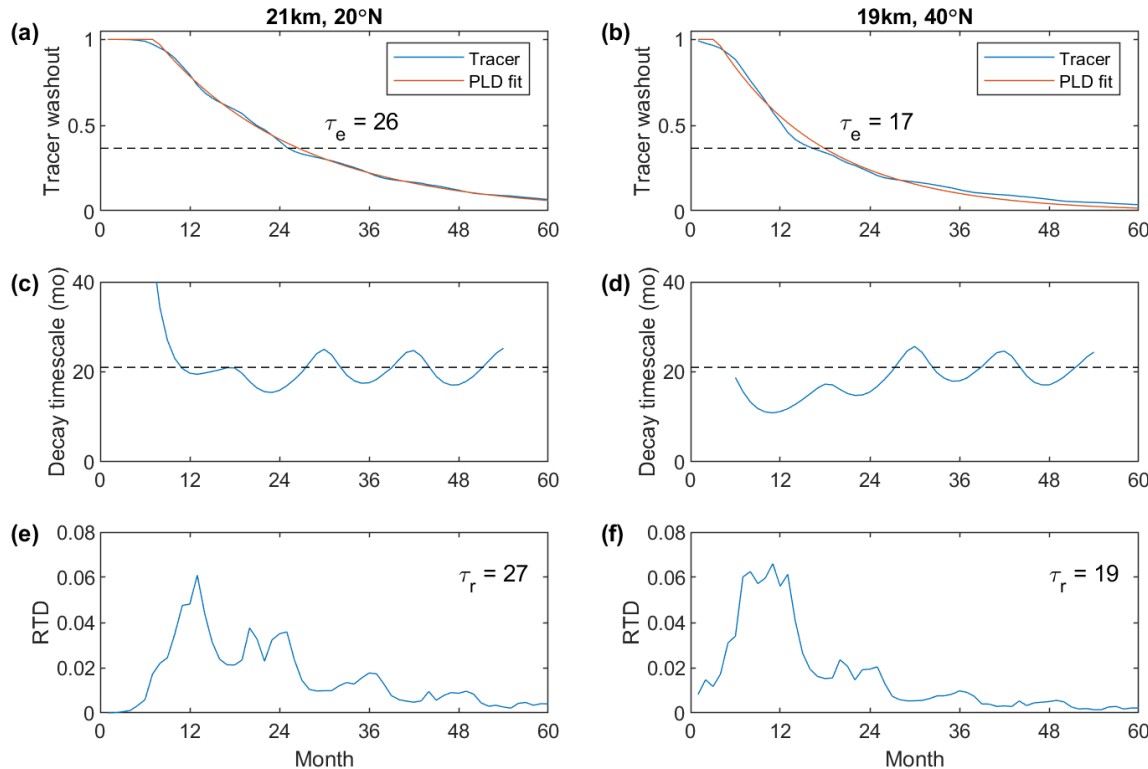


**Figure 3: Diagnostics of simulated tracer stratospheric persistence from two example pulse experiments. (a,b) Time series of stratospheric tracer fraction. Dashed black line indicates the value of 1/e, and $\tau_e$ gives the e-folding time when the stratospheric fraction crosses this threshold. Fits to the simulated washout functions produced using the lagged-decay model of Eq. 5 are shown in red. (c,d) Decay timescale calculated as the running 11-month running exponential fit to the stratospheric fraction time series.**
**(e,f) Residence time distribution, calculated from the washout functions by Eq. 3. Mean residence time $\tau_r$ is calculated from the residence time distribution assuming an exponential decay after 60 months with timescale 20 months.**

Residence time distributions are calculated from the washout functions and shown in Fig. 3e and f. The RTD shows a peak at roughly 13 months for the 21 km, 20°N injection and a broader peak from 7 to 13 months for the 19 km, 40°N injection. Both





cases exhibit a tail on the RTD to larger residence time, with a seasonal variation showing local peaks in stratospheric removal

during NH winter. Mean residence times ($\tau_r$) are calculated from the RTDs using Eq. 1, resulting in values of 27 and 19 months for the two cases. Mean residence time calculations take into account the tail in the RTD beyond the length of the simulations by assuming a continued exponential decay with a timescale of 20 months—for the two examples shown here this increases the calculated mean residence times by 17 to 21 percent. In the two cases shown here, the mean residence time is found to be similar to the e-folding time measured from the time of tracer injection.

For each passive tracer pulse experiment spanning 0-60°N and stratospheric heights from 13 to 25 km, mean residence time results are shown in Fig 4. Mean residence times range from around 2 months in the lowermost extratropical stratosphere (13 km, 50°N for June injection) to over 40 months in the tropical mid-stratosphere ($z \geq 23\,\text{km}, \phi = 0°N$ for both injection months). In the latitude-altitude plane, mean residence time shows a clear dependence on injection altitude, increasing monotonically with injection height with only one exception (at 20°N, 19-21 km for June injection). The vertical gradient of

mean residence time is especially strong in the lower tropical stratosphere, where we find a 3-4 fold increase in residence time for injections between 17 and 21 km. For a given altitude, mean residence times are shortest in the high latitudes, and longest in the tropics. The latitudinal dependence is stronger for summer injections. Weaker sensitivity of mean residence time to injection latitude for winter injections likely arises from the effects of strong mixing (in the "surf zone" (McIntyre and Palmer, 1983)) related to Rossby-wave breaking. Mean residence time for injections into the lowermost stratosphere ($z < 17\,\text{km}, \phi \geq$

40°N) are rather short (2- 10 months).

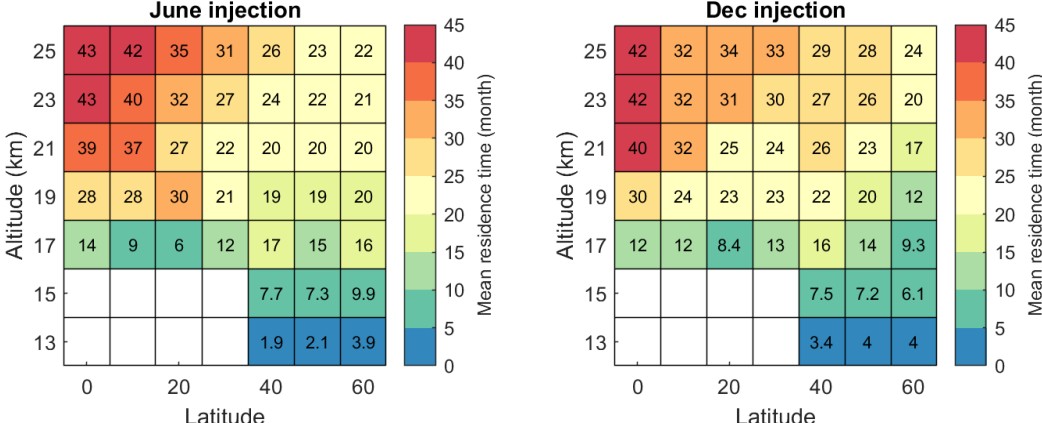

**Figure 4: Stratospheric mean residence times from passive tracer pulse experiments with tracer injection spanning various latitude and height combinations. Results shown for injections in boreal summer (left) and winter (right).**


A clear feature of the tracer washout timeseries shown in Fig. 3 is that for both cases shown, the tracer fraction in the stratosphere remains around 1 for some initial period until decay begins to an appreciable amount. This is also visualized in Fig. 2, which shows how for an equatorial high-altitude injection, it takes some months for the main cloud of tracers to be



transported into the extratropical lower stratosphere where they begin to cross the tropopause into the troposphere. Clearly,

assuming that the stratosphere is well-mixed is not a good assumption for some period after an injection, and thus the decay of tracers from the stratosphere would not be expected to follow an exponential decay for all injection locations.

Figure 3 (a,b) includes nonlinear best-fits using the lagged-decay model described by Eq. 5. For both cases shown, the lagged-decay model clearly produces a reasonable fit to the washout timeseries. This simple model allows us to decompose the total residence time into timescales representing the lag and decay: for example, for the 21 km, 20°N injection case, the best fit is

achieved with a lag timescale of 7.5 months and a decay timescale of 19 months, which sum to give the overall mean residence time of 27 months.

Figure 5 displays the dependence of the lag and decay timescales on injection latitude and altitude for the June passive tracer pulse experiments. Lag timescale is quite small (< 3 months) at altitudes of 17 km and below, and increases rapidly with height, especially in the tropics where values reach 21 months. The lag can be understood to be strongly affected by the

isolation of tracers within the tropical pipe for a tropical injection. For a high altitude, equatorial injection, the lag timescale is almost half of the overall mean residence time. The decay timescale varies most strongly with latitude, with larger values (24-27 months) for tropical injections above 17 km compared to around 12 months for 60°N injections. Decay timescale shows a weaker altitudinal dependence than the lag time, and is larger for tropical injections at 19-21 km than higher altitudes. The decay timescale therefore seems to be related to the "distance" the injection is away from the extratropical lower stratosphere

exit region following the BDC.

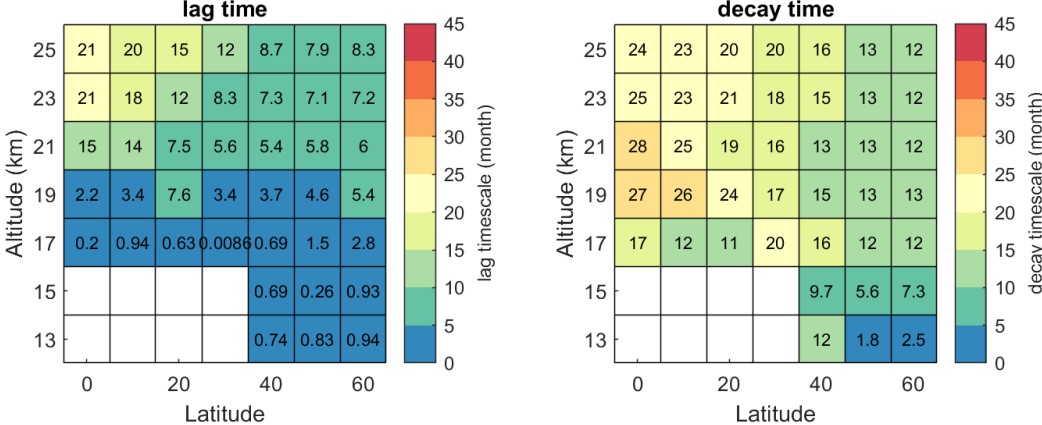

**Figure 5: Best fit lag and decay timescales from application of the lagged-decay model (Eq. 5) to the passive tracer pulse experiments for June injections as a function of injection latitude and altitude.**

**4.2 Observations– stratospheric aerosol lifetime**

Stratospheric aerosol mass anomalies after the June 1991 Pinatubo eruption derived from the GloSSAC multi-spectral measurements are shown in Figure 6a. These mass estimates are qualitatively similar to past estimates based on SAGE II observations (Sukhodolov et al., 2018) but around 20-40% larger, likely because the updated GloSSAC source data





incorporates measurements from multiple instruments, filling gaps in the pure SAGE II record. Stratospheric aerosol mass
increases over the first 5 months after the eruption, traces a plateau between months 5 and ~15, and then decays in an
approximately exponential fashion. The magnitude of the mass plateau (calculated as a mean over the 6-12 month after eruption
period) depends on the width of the unimodal size distribution assumed in the Mie calculations (see Methods), ranging from
approximately 4.7-5.8 TgS (Table 1). These values are roughly consistent with the ~5 TgS total stratospheric sulfur injection
estimated by Ukhov et al. (2023). Total column aerosol mass estimates from the HIRS instrument were derived by Baran and
Foot (1994) and are compared to the GloSSAC-drived values in Fig. 6a. Between months 6 and 20 after the eruption, the HIRS
time series follows a similar shape as the GloSSAC time series, albeit with a smaller amplitude. The total column aerosol mass
cannot be smaller than the stratospheric component, so clearly there is systematic error involved: the HIRS estimates might be
too small through this period or the GloSSAC estimates too large—or both. However, the HIRS estimates are not strongly
different from GloSSAC data using the $\sigma = 1.2$ assumption, with a mean difference of 0.35 TgS, or 8% over this period. In
the first 3 months after the eruption, the HIRS mass estimate is larger than the GloSSAC-based estimates by up to 2-3 TgS.
This difference is roughly consistent with the amount of Pinatubo's total sulfur emission we might expect to be in the
troposphere based on the total emission of 7.5 TgS (Carn, 2022) and the estimate of 1/3 of the total emission being tropospheric
(Ukhov et al., 2023).

The simplest measure of aerosol persistence is the e-folding time, calculated here as the number of months after the eruption
when the aerosol mass crosses $1/e$ of its maximum value. For the GloSSAC-derived mass time series, we use the plateau
average as the peak value, resulting in e-folding times of approximately 25 months (Table 1). For the HIRS time series, we
use the peak value at 3 months after the eruption as the maximum, and the resulting e-folding time is 20.4 months.

When aerosol mass decreases after the plateau period, the decay timescale for all three GloSSAC mass time series fairly
quickly reaches a values of between 13 and 6 months, with a mean value of 9.6 over the 24-48 month period (Fig. 6b). That
this value is significantly shorter than the decay timescale in the tracer simulations illustrates the impact that gravitational
settling has on the decay of stratospheric aerosol. The decay timescale is found to be insensitive to assumptions regarding the
aerosol size distribution. The HIRS data set decays more rapidly than the GloSSAC time series starting at 21 months. As a
result, the HIRS decay timescale is shorter during the main decay period.




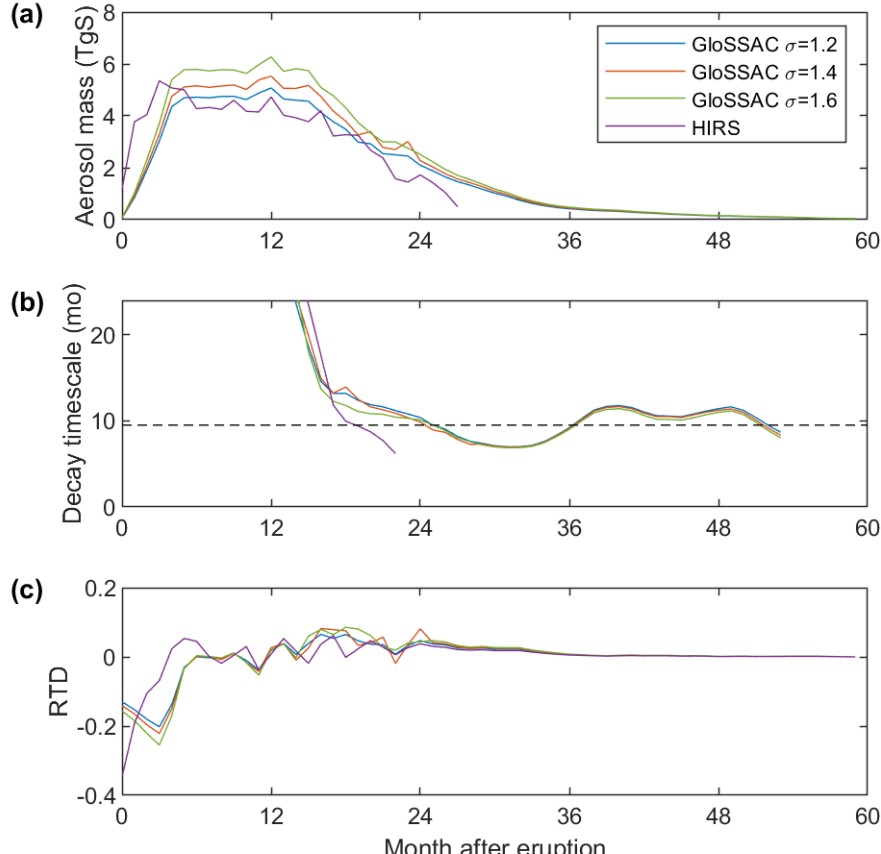

**Figure 6: Diagnosing the lifetime of stratospheric aerosol from the 1991 Pinatubo eruption. (a) Three estimates of global stratospheric aerosol mass derived from the GloSSAC data set with different assumptions of the aerosol size distribution, and one estimate of near-global (80°S-80°N) total column aerosol mass from HIRS. (b) Decay timescale for the aerosol mass time series, calculated as a running 11-month exponential fit. (c) Residence time distributions derived from the aerosol mass time series.**


**Table 1: Aerosol persistence metrics derived from Pinatubo aerosol observations.**

| Global aerosol mass data | Peak aerosol mass (TgS) | e-folding time (months) | Mean lifetime (months) |
| --- | --- | --- | --- |
| GloSSAC ($\sigma = 1.2$) | 4.7 | 25 | 22 |
| GloSSAC ($\sigma = 1.4$) | 5.2 | 25 | 23 |
| GloSSAC ($\sigma = 1.6$) | 5.8 | 25 | 22 |
| HIRS* | 5.4 | 20 | 15 (11-20) |




The aerosol mass time series can be used to determine a residence time distribution and mean lifetime if they are first converted to a washout function as described in Sect. 2. This requires normalizing the mass time series by the total amount of aerosol mass removed from the stratosphere. Given the relatively flat plateaus in the GloSSAC-derived mass timeseries, we assume that all injected sulfur has been converted to aerosol before the removal begins, so that the plateau mass value is representative

of the total mass of sulfur injected (and so also of that removed). Under this assumption, the residence time distributions are very similar for the three GloSSAC-derived mass timeseries (Fig. 6c), each showing a main peak of removal at around 16-19 months, and negative values in the first months representing the production of aerosol. Mean lifetime is relatively insensitive to the assumed width of the size distribution, with values of around 22 months resulting from each of the GLoSSAC timeseries (Table 1). The mean lifetimes are around 2.5 months smaller than the e-folding times since they incorporate the effect of the

production of aerosol in the first months. To calculate a lifetime from the HIRS total column mass time series, which covers only the first 27 months after the Pinatubo eruption, we first extend the time series by concatenating it with the GloSSAC $\sigma = 1.2$ timeseries from 21 months where the two time series overlap, and produce a washout function by normalizing by the total sulfur injection value of $7.5 \pm 2.3$ TgS (Carn, 2022), resulting in a mean residence time of 15 months with a range of 11-21 months. The HIRS-derived lifetime is thus much shorter than that derived from GloSSAC, which is consistent with the idea

that the HIRS-derived mass represents both tropospheric and stratospheric aerosol, and that the tropospheric component is rapidly removed in the first weeks after the eruption.

The ~22-month lifetime of stratospheric aerosol derived here is based directly on the GloSSAC data product, including the assumption that the peak values are representative of the total sulfur injected. The result also clearly depends on the accuracy of the mass time series: if for example the width of the aerosol size distribution changes with time, this could perturb the shape

of the washout function which would affect the mean lifetime. Also, if the lack of SAGE-II observations in the lower tropical stratosphere in the first few months were to result in an underestimate of the aerosol mass, our estimate would be biased as a result.

To explore how uncertainties in the observation-based stratospheric aerosol mass timeseries would affect the aerosol lifetime results, we first construct a rough confidence interval within which we surmise the true timeseries of aerosol mass is likely to

be within. The construction of this interval is subjective, we only wish to attempt to define an interval within which we can credibly assume the correct answer lies within. As a lower bound, we take the $\sigma = 1.2$ GloSSAC time series and subtract 0.35 TgS—a value about 7% of the peak mass—to account for potential biases in the GloSSAC data and our conversion to aerosol mass. For an upper bound, we entertain the possibility that the HIRS observed aerosol mass is representative of a stratospheric amount, but that the HIRS timeseries is low-biased. We thus scale HIRS up so that it is consistent with the $\sigma = 1.4$ GloSSAC

timeseries over the plateau period, then use either the scaled HIRS values or the $\sigma = 1.6$ GloSSAC timeseries, which ever is greater at any point in time, and finally add an additional offset of 0.35 TgS to produce the upper bound. The resulting range is shown in Figure 7a.





**Table 2: Parameters of the production-lag-decay model varied in the Monte Carlo experiments to explore the effect of aerosol mass uncertainty on aerosol lifetime, and their lower and upper bounds.**

| Parameter | Lower bound | Upper bound |
|---|---|---|
| Stratospheric sulfur injection ($M_{SO2}$, TgS) | 4 | 10 |
| Production time scale, box 1 ($\tau_{p,1}$, months) | 1 | 3 |
| Lag timescale, box 1 ($\tau_{l,1}$, months) | 12 | 20 |
| Decay timescale, box 1 ($\tau_{d,1}$, months) | 8 | 12 |
| Production time scale, box 2 ($\tau_{p,2}$, months) | 0 | 1 |
| Decay timescale, box 2 ($\tau_{d,2}$, months) | 1 | 6 |
| Fraction of injection in box 1 ($\epsilon$) | 0.5 | 1 |

To produce hypothetical mass time series falling within these bounds and translate the time series into aerosol lifetimes, we utilize the simple production-lag-decay (PLD) model (Eq. 6). We use a Monte-Carlo technique, constructing a large number of hypothetical timeseries using ranges of values for the injected sulfur amount and the three model parameter values (Table 2). Constructed mass timeseries that fall outside our confidence interval are discarded.

A small sample of hypothetical aerosol mass timeseries are shown in Fig 7a, and a histogram of the lifetimes of the set of time series that fall within the confidence interval is shown in Fig 7b. The distribution peaks at 23-24 months, and has a mean value of 23.5 months.

Another scenario that could be envisaged is that a portion of aerosol is produced and removed quickly from the stratosphere without being observed by the instruments used to construct GloSSAC, for example in the tropical lower stratosphere. To test the potential impact of this on the resulting aerosol lifetime, we perform a further test, which uses a two-box implementation of the PLD model. One box represents the relatively long-lived aerosol consistent with the GloSSAC mass timeseries, and a second box represents a short-lived population of stratospheric aerosol that might be present and removed within the first few months. The resulting mass timeseries could have a "bump" in the first months, similar to that seen in the HIRS time series. The PLD two-box model takes eight parameters (Table 2), the mass injected into each box and the 3 model parameters for each box: we reduce this to 7 by assuming zero lag in the short-lived box, as suggested by the small lag timescales seen in our tracer experiments for lower stratospheric injections (Fig. 5). Some sample time series are shown in Fig. 7c superimposed on the constructed confidence range to illustrate the type of evolutions possible. Under this assumed model, the distribution of resulting overall stratospheric aerosol lifetimes is shown in Fig. 7d. The two-box model allows for solutions with smaller lifetimes than the single box model—the resulting distribution of time series which lie within the uncertainty range are associated with lifetimes that span 12-26 months, with a broad peak centered around 19 months. A bivariate histogram of the two-box model results (Fig. 8) shows how stratospheric aerosol lifetimes decreases with an increasing amount total amount of



stratospheric sulfur injection assumed. Based on the $5.0 \pm 1.5$ TgS estimate of stratospheric sulfur injection, we take 6.5 TgS as a $1\sigma$ upper bound. A histogram of lifetime for time series with injection less than 6.5 TgS is shown in Fig. 7d in red: this

distribution constrained by the range of most likely injection amounts shows a peak at 21-22 months, and a minimum of 18 months. We interpret these results as suggestive that lifetimes below 18 months are unlikely given the observed aerosol mass timeseries and sulfur injection amounts and the uncertainties on these observations.

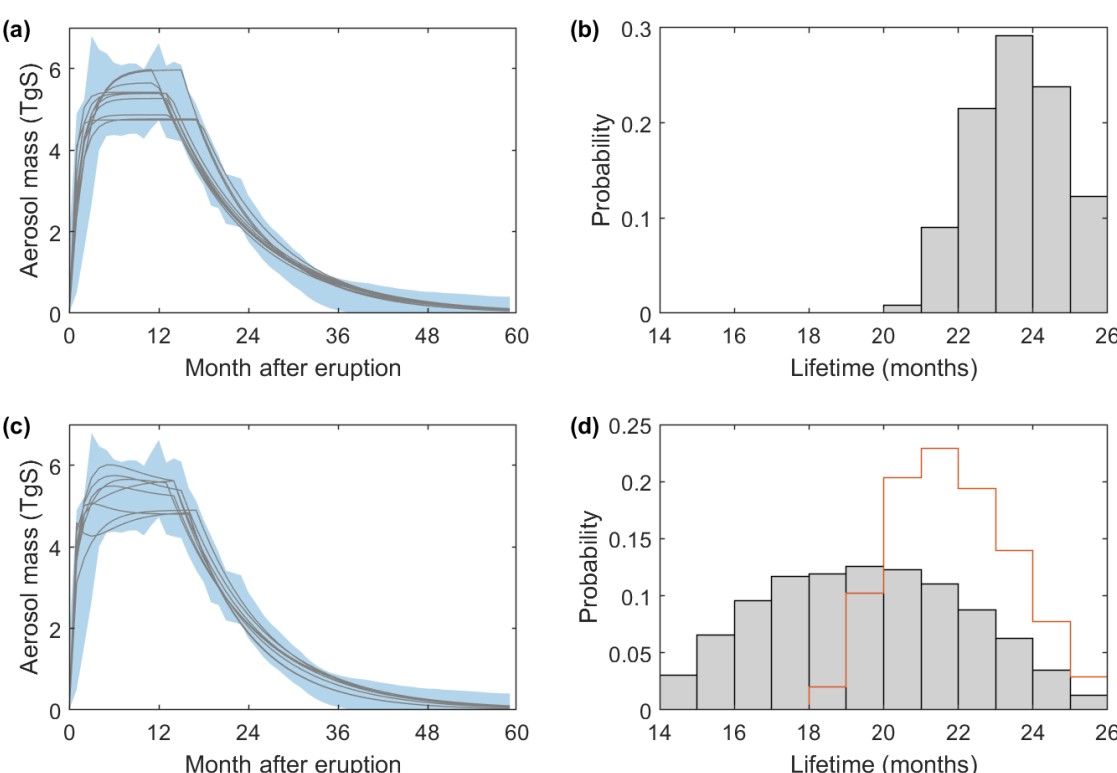


**Figure 7: Modeling the impact of stratospheric aerosol mass uncertainty on aerosol lifetime. (a) Constructed aerosol mass uncertainty range (blue) and a sample of hypothetical mass timeseries generated with the production-lag-decay model (gray) that fit within the uncertainty range. (b) Histogram of aerosol lifetimes corresponding to the full set of hypothetical time series consistent with the uncertainty range. (c) As (a), but with hypothetical time series constructed using a two box PLD model. Time series shown**
**in gray are a sample of those with lifetimes between 12 and 18 months. (d) As (b) but for the two-box PLD model results. Histogram for hypothetical mass time series for injections less than 6.5 TgS shown in red.**



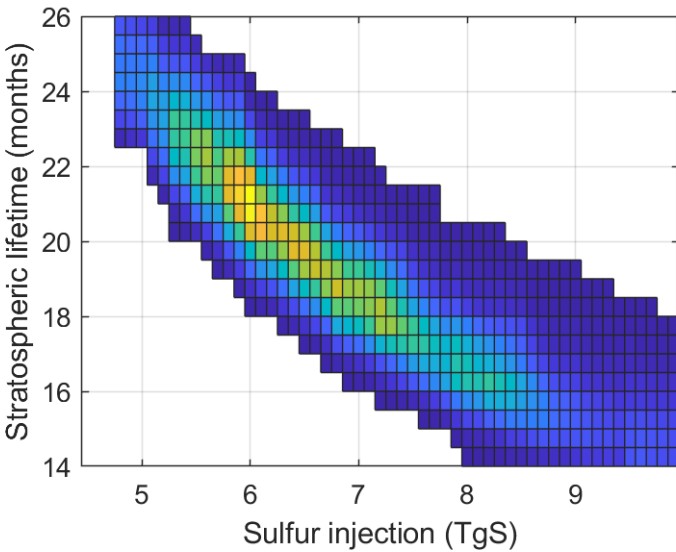

**Figure 8: Bivariate histogram showing the distribution of lifetime of hypothetical aerosol time series as a function of the total**
**stratospheric sulfur injection in two-box PLD output constrained by the uncertainty range based on observations of aerosol from the 1991 Pinatubo eruption.**

## 5. Conclusions

A primary concrete conclusion of this work is that the lifetime of stratospheric aerosol from the Pinatubo eruption based on
the GloSSAC global aerosol reconstruction is around 22 months, significantly longer than the 12-month residence time (or lifetime) widely quoted. This difference results in large part because our estimate includes the roughly 12–15-month lag between the eruption and the initiation of the decay of stratospheric aerosol, in contrast to other studies which have implicitly assumed that the decay timescale is equivalent to a residence time. Examination of the observed spread of aerosol after Pinatubo and passive tracer experiments supports the idea that a majority of the aerosol from the eruption took some time to
be transported from the tropical stratosphere to the extratropical lower stratosphere where removal from the stratosphere occurs. Thus, while our investigation confirms the roughly 10–12-month decay timescale of the Pinatubo aerosol, the lifetime of stratospheric aerosol must include the lag between injection and initiation of removal, and so the lifetime is best described as being around 22 months. Our results also show that the calculated lifetime is roughly similar to the e-folding time measured from the eruption date, showing that a relatively easy to estimate quantity can be used to provide a good estimate of aerosol
lifetime, at least for large tropical eruptions.

Our estimate of the Pinatubo aerosol lifetime is strongly tied to the accuracy of the GloSSAC data set. The most important source of uncertainty in the observations in terms of calculating a mean aerosol lifetime is the possibility of the GloSSAC



observations underestimating the aerosol extinction—and therefore the amount of sulfate aerosol—in the first 18 months when SAGE II extinction measurements are missing in the lower stratosphere, especially in the tropics. We find that if a substantial amount of the initial $SO_2$ injection was converted to aerosol particles which were not observed by the space- and ground-based instruments used in GloSSAC and quickly removed from the stratosphere, then the mean aerosol lifetime could be less than the 22 months estimated here.

The key consideration for the effect of observational uncertainties on the mean aerosol lifetime is the total mass of the initial injection. The total $SO_2$ emitted from Pinatubo is estimated to be $7.5 \pm 2$ TgS (Carn, 2022), and our estimates of sulfur aerosol mass based on GloSSAC extinctions peak at between roughly 5 and 6 TgS depending on size distribution assumptions. Notably, our peak mass estimates are broadly similar to the $5.0 \pm 1.5$ TgS stratospheric injection amount obtained by combining the total emission from Carn (2022) with the inverse modelling results of Ukhov et al. (2023). This consistency strengthens the argument that there is no gross underestimation in the GloSSAC mass time series, and that the lifetime calculation based directly on the mass time series is reasonably accurate. On the other hand, given the rapid removal we would expect for injection into the very lowest tropical stratosphere based on passive tracer simulations, and the likelihood that the sulfur injection profile for Pinatubo extended through the stratosphere up to the plume height observed, it seems quite possible that some amount of aerosol was produced in the lower stratosphere and quickly removed. This would decrease the mean lifetime, to a degree that depends on the amount of injected sulfur quickly removed as aerosol. We have explored the potential impact of rapid removal through Monte Carlo simulations using our production-lag-decay model, which suggest that for a maximum total stratospheric injection of 6.5 TgS, a mean lifetime of less than 18 months is very unlikely.

Our passive tracer experiments illustrate how stratospheric residence time—which along with gravitational settling velocity plays a role in the lifetime of stratospheric aerosol—depends strongly on the altitude and latitude of the tracer injection. Our results are qualitatively consistent with the results of Sun et al. (2023), who simulated aerosol injections at various heights between 16 and 24 km between 30°S and 30°N in different seasons and found generally longest lifetimes for injections near the equator. As our experiments were largely focused on the 1991 Pinatubo eruption, the tracer pulse experiments are based on injections only in June and December 1991, and so do not assess interannual variability in stratospheric dynamics, e.g., that due to the quasi-biennial oscillation (Trepte and Hitchman, 1992; Pitari et al., 2016; Visioni et al., 2018). Model simulations suggest that aerosol persistence can also be sensitive to the particular meteorological conditions at the time of eruption (Quaglia et al., 2023; Zhuo et al., 2024; Fuglestvedt et al., 2024). Furthermore, there is the possibility that the residence times estimated through our pulse experiments are affected by changes in stratospheric dynamics brought about by heating of the Pinatubo aerosols, and so may be to some degree inaccurate for injections to locations other than that of the Pinatubo aerosol. We assume that the general features of our analysis are indicative of characteristics of the dependence on altitude and latitude, but certainly more extensive study could confirm these results and assess the importance of interannual and meteorological variability.

The passive tracer pulse experiments suggest that stratospheric residence time is strongly dependent on the altitude of injection, especially in the first 4 km above the tropical tropopause where residence time increases by a factor of 4. This implies that rather than the tropopause acting as a binary threshold controlling the climate impact of eruptions (Aubry et al., 2016), the





region between 17 and 21 km be thought of as a transition region where the climate impact increases strongly with increasing injection height. Pulse injections into the lowermost stratosphere (below 17 km, poleward of 30°) lead to residence times which depend on injection height and latitude, but center roughly around 6 months, roughly consistent with prior estimates for extratropical eruptions (Oman, 2005). Nonetheless, the tracer experiments show a significantly longer residence time (>20 months) for extratropical injections above the level of the tropical tropopause (17 km), supporting the idea that extratropical eruptions with injection heights comparable to observed tropical eruptions like Pinatubo could lead to aerosol with lifetimes long enough to significantly affect hemispheric climate (Toohey et al., 2019).

Stratospheric residence time estimated from the tracer pulse experiments for tropical injections between 19-23 km are clearly much longer than the observed lifetime of Pinatubo aerosol. The difference is almost certainly largely explained by the impact of gravitational settling of the aerosols. This difference is also illustrated by comparing the spread of simulated tracers to observed aerosol in Fig. 2, where we see observed aerosol more confined to lower altitudes, especially after 6 months after eruption. Very roughly, if we take 40 months as a typical value of residence time for tropical injection at the height of the Pinatubo plume (~25 km) from the tracer pulse experiments, and compare that to the estimated 22 month lifetime of the aerosol, we can estimate that gravitational settling reduces the residence time by around 18 months or 45%.

Implications of our work extend beyond understanding of the lifetime of the aerosol from Pinatubo eruption. The framework developed here for describing the temporal evolution of stratospheric aerosol in terms of timescales for the production, lag and decay will be useful in simple models used to generate volcanic aerosol forcing fields for climate models (Aubry et al., 2020; Toohey et al., 2016). The same framework may also prove useful for comparing the very different time evolutions of aerosol in comprehensive aerosol-climate models used in simulations of volcanic eruptions (Timmreck et al., 2018; Clyne et al., 2021; Quaglia et al., 2023), since it appears that while models generally produce similar aerosol decay timescales, the lag between injection and decay initiation is strongly model dependent (Quaglia et al., 2023). Similarly, the framework is likely to aid in the interpretation of simulated geoengineering scenarios (Visioni et al., 2018; Sun et al., 2024). A better understanding of the rate of removal of aerosol from the stratosphere may prove useful in estimating the dates of unidentified eruptions from the sulfur deposited to polar ice sheets (e.g., Toohey and Sigl, 2017; Sigl et al., 2022). Finally, residence times estimated through passive tracer pulse experiments may be directly applicable to volcanic gas emissions which remain in the gas phase, e.g., the water vapor injection from the 2022 Hunga Tonga eruption (Millán et al., 2022).

**Appendix A: Derivation of a washout function for a secondary tracer**

For a tracer injected into a reservoir at time zero, following the lagged decay model introduced in the main text, the washout function (the normalized amount of tracer in the reservoir) is:

$$W = \begin{cases} 1, & t < \tau_l \\ \exp\left(\dfrac{-(t - \tau_l)}{\tau_d}\right), & t \geq \tau_l \end{cases}.$$



Consider the case that the tracer injected is initially in one state (as the primary tracer) and is converted to another state (secondary tracer) with a timescale $\tau_p$. In the present context, we are thinking about the conversion of gaseous $SO_2$ to sulfate aerosol, but the situation is generalized to any chemical or physical change in the tracer. The fraction of tracer in the secondary state is

$$f = 1 - \exp(-t/\tau_p).$$

The product $fW$ represents the fraction of the initial tracer injection in the secondary tracer state in the reservoir as a function of time. This is not however a washout function for the secondary tracer, since the washout function must represent the fraction of tracer at any time to the total amount of tracer that spends time in the reservoir. If some of the primary tracer is removed from the reservoir before conversion to secondary tracer, then the product $fW$ needs to be normalized in order to properly represent a washout function for the secondary tracer.

The rate of removal of total tracer (i.e., primary plus secondary) from the reservoir can be derived from the washout function, it is $-dW/dt$, and the rate of removal of the secondary tracer is this total removal rate times the fraction of tracer in the secondary state:

$$R_2 = -f\left(\frac{dW}{dt}\right) = \begin{cases} 0 & ,t < \tau_l \\ \left[1 - \exp\left(-\frac{t}{\tau_p}\right)\right]\left[\frac{1}{\tau_d}\exp\left(\frac{-(t-\tau_l)}{\tau_d}\right)\right], & t \geq \tau_l \end{cases}.$$

The total amount of tracer removed as secondary tracer is the integral of the removal rate above, i.e.:

$$F_2 = \int_{\tau_l}^{\infty}\left[1 - \exp\left(-\frac{t}{\tau_p}\right)\right]\left[\frac{1}{\tau_d}\exp\left(\frac{-(t-\tau_l)}{\tau_d}\right)\right]dt.$$

This integral can be solved, resulting in an expression for the amount tracer removed as secondary tracer:

$$F_2 = 1 - \frac{\tau_p}{\tau_d+\tau_p}\exp\left(-\frac{\tau_l}{\tau_p}\right).$$

The washout function for the secondary tracer is thus the total tracer washout function $W$, multiplied by the fraction of tracer in the secondary form $f$, divided by the total amount of tracer to be removed as secondary tracer $F_2$:

$$W_2 = \begin{cases} \frac{1}{F_2}\left[1 - \exp\left(-\frac{t}{\tau_p}\right)\right], & t < \tau_l \\ \frac{1}{F_2}\left[1 - \exp\left(-\frac{t}{\tau_p}\right)\right]\exp\left(-\frac{t-\tau_l}{\tau_d}\right), & t \geq \tau_l \end{cases}$$

## Appendix B: Considering Cerro Hudson

$SO_2$ emission estimates for Pinatubo are around 7.5 TgS, while for the August 1991 eruption of Cerro Hudson (46°S) the estimate is 1.3 TgS (Carn, 2022), suggesting the aerosol mass in the following months could be up to 18% from the Cerro Hudson eruption. Although the eruption height for Cerro Hudson is estimated by (Carn, 2022) to reach 18 km, visual inspection



of GloSSAC extinction in August 1991 (Fig. B1) shows a clear enhancement centered at around 12 km and relatively contained to latitudes poleward of 60°S, which strengthens in September and persists for the next months. The position of this aerosol enhancement is consistent with observations of the $SO_2$ cloud, which note a southward transport of the relatively coherent cloud in the days following the eruption (Doiron et al., 1991). Aerosol between 20 and 25 km in August shows a gradient towards the tropics, strongly suggesting this enhancement is Pinatubo aerosol mixing out of the tropics into the mid-latitudes. In order to approximately quantify the contribution of Cerro Hudson to the global stratospheric aerosol mass in the months after the two eruptions, we assume that Cerro Hudson's aerosol was mostly confined to the SH lower-most stratosphere, i.e., $z < 17$ km and $\phi < -30$. Five months after Pinatubo, and 3 months after the Cerro Hudson eruption, which is when large quantities of aerosol from the tropics is seen to start entering the SH LMS, the aerosol mass in the SH LMS is around 0.25 TgG, around 5% of the global aerosol mass. As the Pinatubo aerosol enters the SH LMS it is difficult if not impossible to separate the contributions of the two eruptions, and we do not attempt to do so here. Since the apparent contribution of Cerro Hudson to the global aerosol mass in the years 1991 and after is rather small, for simplicity we treat the observed aerosol as if it is entirely from the Pinatubo eruption.

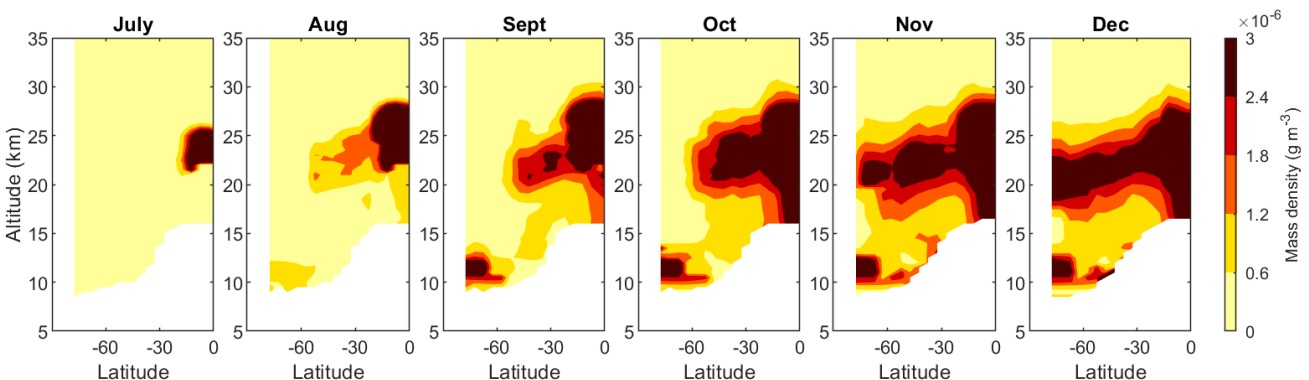

**Figure B1: Monthly mean zonal mean Southern Hemisphere aerosol mass density for July-Dec 1991 derived from the GloSSAC multi-spectral extinction data set.**

**Author Contribution**

MT, ST and YJ designed the tracer experiments and YJ performed the simulations. SK derived aerosol mass from the GloSSAC dataset. MT led the analysis, and wrote the manuscript with input from all co-authors.

**Competing Interests**

At least one of the (co-)authors is a member of the editorial board of Atmospheric Chemistry and Physics.



**Acknowledgements**

MT and SK acknowledge the support of the Canadian Space Agency through grant 21SUASOCSA.

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
