# Peer review of "Stratospheric residence time and the lifetime of volcanic stratospheric aerosols"

_EGUsphere, 2024_

## Author Response (AR1)

Dear editor,

We thank the reviewers for their positive appraisals of the submitted manuscript, and their edit suggestions, which we are confident have helped us improve the paper.

Below, the reviewer comments are reproduced below in black, and our responses in blue.

Reviewer 1 (Daniele Visioni)

This manuscript provides a very interesting perspective on the issue of the persistence of stratospheric aerosols through residence time, by developing a new framework to better understand (and potentially separate) the combined effect of stratospheric circulation and aerosol microphysics when quantifying the lifetime of volcanic (or potentially of SAI) aerosols. The manuscript is of extremely high value and quality, and I fully endorse publication. I have a few very minor comments and notes that the authors might wish to consider, but the manuscript is essentially publishable in its present form.

I want to add that I particularly liked the clear explanations in Section 2, which make this a very valuable "entry-point" manuscript for researchers interested in this topic. The analyses in Section 3 are also very robust and thorough.

On Figure 4, I would also note the consistency in the results with the experiments shown in Visioni et al. (2019): in that case, injections of $SO_2$ at 24 km at different latitudes and seasons were also tested in CESM-WACCM (for SAI simulations purposes, so more a three-months injection than a pulse) but there is a very robust similarity between the results (here, measured in terms of lifetime, there, in terms of burden) showing higher residence times for June than for December.

We agree, these results are consistent and he have added a description of this in the conclusions section where we compare our results to prior studies.

In Section 4.2 I would suggest that when the authors say "the HIRS estimates might be too small through this period or the GloSSAC estimates too large" they acknowledge that, for instance, the Quaglia et al. (2023) manuscript elsewhere cited point to the fact that the data [...] "are affected by a systematic error of 10 % due to the sensitivity of the retrieved method and uncertainties in the background.".

Thank you, good point. We have added a statement to point out that the difference between the HIRS mass timeseries and our GloSSAC based estimate (using sigma=1.2) is comparable to the 10% systematic error estimate of HIRS data reported by Baran and Foot (1996).

Finally, I will note in the conclusions that the framework here developed would be of great use not just to " aid in the interpretation of simulated geoengineering scenarios", but also in the development of more robust climatic emulators for SAI. In particular, there are now some efforts such as Farley at al (2024) to develop climatic emulators to simulate a broader range of interruptions of terminations of stratospheric aerosol injections, and adding a more robust representation of residency time could greatly improve our understanding (i.e. see their figure 6).

Yes, we agree the framework could be useful in SAI emulation work and have added a statement and reference to Farley et a. (2024) in the conclusions section as suggested.

References

Farley et al 2024 Environ. Res.: Climate 3 035012 DOI 10.1088/2752-5295/ad519c

Visioni, D., MacMartin, D. G., Kravitz, B., Tilmes, S., Mills, M. J., Richter, J. H., & Boudreau, M. P. (2019). Seasonal injection strategies for stratospheric aerosol geoengineering. *Geophysical Research Letters*, 46, 7790–7799. https://doi.org/10.1029/2019GL083680

**Anonymous Referee #2**

The manuscript addresses an important and timely question of aerosols' lifetime in the stratosphere. Authors mostly focus on the Pinatubo case, as it is the largest relatively well observed volcanic case, but the implications go well beyond this specific case, given a growing scientific interest in potential geoengineering schemes through stratospheric aerosol injections. Authors conceptualize the topic through basic formulation and simple modelling and bring some order to the existing in the literature variety of metrics and estimates. The FLEXPART modelling is then used to demonstrate the dependence of the mean residence time on the altitude and latitude of injections. Contrasting the FLEXPART and Monte-Carlo PLD modelling with the Pinatubo observationally-derived data additionally allows to estimate the approximate impact of gravitational sedimentation and the range of mean aerosol lifetime after Pinatubo. The paper is very well written and inspires further research looking at the physics and dynamics of aerosol transport. I have only minor comments listed below:

L40: You can also mention Quaglia et al. (2023), who presents e-folding time estimates for GloSSAC (AOD and burden), AVHRR (AOD), HIRS (burden), and for 6 models

Indeed, we think the e-folding times reported by Quaglia et al. (2023) and other studies are important to the discussion, but at this particular point in the manuscript we are concerned with explaining the origin of the oft-quoted "12-month lifetime" for stratospheric aerosol from a large tropical eruption, so we focus on the observational studies upon which this interpretation is built. Based on this comment, we have slightly expanded the description of the relevance of the study of Quaglia et al. (2023) and other related studies towards the end of the introduction as motivation for our study.

L50: It is worth mentioning here also the self-lofting process (e.g., Khaykin et al., 2022). Sukhodolov et al. (2018) also performed a Pinatubo experiment without aerosol effects on temperature, which resulted in a faster burden decay.

Thank you, this is a very good point. We have added 2 sentences in the discussion of processes affecting aerosol persistence with reference to self-lofting and the references mentioned.

Figure 1 caption: Please indicate what are the dashed black lines in panels (b) and (c) (e-folding time?)

Thanks, we have added a sentence in the figure caption to explain the black dashed line as the value $e^{-1} = 0.368$, from which the e-folding time can be determined.

L212: Note that CMIP6 aerosol forcing data (ftp://iacftp.ethz.ch/pub_read/luo/CMIP6/, Revell et al., 2017) also contained aerosol mass and effective radius inversed from the available wavelengths of GloSSAC. It doesn't mean it is the most correct timeseries, as it likely also used an older version of GloSSAC, but consider using it for the intercomparison with your estimates, given that it has been widely used for the recent Pinatubo MIP study by Quaglia et al. (2023)

Good point. For simplicity, we prefer to not display the CMIP6 mass on our plots, since our analysis provides some uncertainty in the mass timeseries by providing estimates using different widths of the size distribution and we prefer to keep the plots as simple as possible. However, we agree it is quite useful to point out that our estimates are quite comparable to the CMIP6 mass data, and we have added statements to point this out in the results section, including a quantitative comparison of our peak mass values (4.7-5.8 TgS, depending on assumed distribution width to the 5.0 TgS sulfur peak mass loading quoted by Quaglia et al (2023)).

L244: Maybe remind here that it is the June 15 case, given that you have two.

Good point, done.

L284: repetition of "running"

Thanks, fixed!

L277: Sun et al. also included sedimentation in their calculations, which they also noted as an important limitation for the transport, while in your case it is not included. Why would it be so close then?

Yes, this is a good point. It is certainly possible that these results agree "for the wrong reasons" i.e., because of compensating differences between the two model simulations. On the other hand, we suggest that it is also possible that the effect of gravitational settling might be rather small in the study of Sun et al. since their prescribed aerosol radius ($r = 0.2$ µm) is rather small. For example, Hamill et al. (1997) provide a simple numerical experiment and estimates it would take a particle with $r = 0.25$ µm 1.3 years to fall from 22 to 20 km, and concludes sedimentation is not an effective removal mechanism, expect for very large particles.

We have modified the text here to point out that the Sun et al. study indeed includes the effect of sedimentation in contrast to our simulations, and also include comparison to the crude ~2-year "turnover time" from the seminal study of Holton et al. (1995). But, since this result (stratospheric residence time for a well-mixed tracer) is not a primary result of our study, we feel it is beyond the scope of the paper to discuss in detail the comparison of our result with that of Sun et al., and suggest this be a motivation for future work.

Figure 2: You discuss the lower row only in the conclusions, while this figure appears in the beginning of the paper. Consider using this part for the analysis earlier

Thanks, good point. We have added a few sentences at the start of Sec. 4.2 to introduce the bottom row of Fig. 2, this actually helps to motivate the use of the PLD model later.

L304: ...lowermost 'extratropical' stratosphere...

Here we mean the "lowermost stratosphere" as defined by Holton et al., (1995), i.e., that part of the stratosphere for which potential temperature surfaces cross the tropopause. We have made that more explicit here, with reference to the Holton et al. paper, and also modified a few other uses of "lowermost" in order to avoid confusion.

Figure 4: This is a very cool plot, clearly showing the importance of gravitational settling, i.e. that it will help aerosols to fight the tropical upwelling and bring them closer to the lower levels, from which they can be more effectively transported out of the tropics by the shallow BDC branch

Thanks!

L339: Ok, I see you intercompare with CMIP6 here. Note that the newer version has been used in Quaglia et al 2023

See our response concerning L212 above, this is where we have made the related changes to the text.

L432: Repetition of "amount"

Fixed, thanks!

L454: lifetime -> mean lifetime

Yes, thanks!

L495: Right, it has been shown that volcanic and geoengineering aerosols can modulate their own transport and disrupt QBO (Nemeier and Schmidt, 2017; Brown et al., 2023; Wunderlin et al., 2024). It is worth adding a couple of citations here on that.

Yes, we agree and have added the suggested citations with direct reference to aerosol impact on the QBO.

We we have also become aware of recent publications (Millán et al., 2024, Zhou et al., 2024; Fleming et al., 2024) on the evolution of stratospheric water vapour after the 2022 Hunga eruption, which our results are very clearly relevant to. We have added brief statements to our conclusions with reference to these new publications.

Brown, F., Marshall, L., Haynes, P. H., Garcia, R. R., Birner, T., and Schmidt, A.: On the magnitude and sensitivity of the quasi-biennial oscillation response to a tropical volcanic eruption, Atmos. Chem. Phys., 23, 5335–5353, https://doi.org/10.5194/acp-23-5335-2023, 2023.

Khaykin, S.M., de Laat, A.T.J., Godin-Beekmann, S. et al. Unexpected self-lofting and dynamical confinement of volcanic plumes: the Raikoke 2019 case. Sci Rep 12, 22409 (2022). https://doi.org/10.1038/s41598-022-27021-0

Niemeier, U. and Schmidt, H.: Changing transport processes in the stratosphere by radiative heating of sulfate aerosols, Atmos. Chem. Phys., 17, 14871–14886, https://doi.org/10.5194/acp-17-14871-2017, 2017.

Revell, L. E., Stenke, A., Luo, B., Kremser, S., Rozanov, E., Sukhodolov, T., and Peter, T.: Impacts of Mt Pinatubo volcanic aerosol on the tropical stratosphere in chemistry–climate model simulations using CCMI and CMIP6 stratospheric aerosol data, Atmos. Chem. Phys., 17, 13139–13150, https://doi.org/10.5194/acp-17-13139-2017, 2017.

Sukhodolov, T., Sheng, J.-X., Feinberg, A., Luo, B.-P., Peter, T., Revell, L., Stenke, A., Weisenstein, D. K., and Rozanov, E.: Stratospheric aerosol evolution after Pinatubo simulated with a coupled size-resolved aerosol–chemistry–climate model, SOCOL-AERv1.0, Geosci. Model Dev., 11, 2633–2647, https://doi.org/10.5194/gmd-11-2633-2018, 2018.

Quaglia, I., Timmreck, C., Niemeier, U., Visioni, D., Pitari, G., Brodowsky, C., Brühl, C., Dhomse, S. S., Franke, H., Laakso, A., Mann, G. W., Rozanov, E., and Sukhodolov, T.: Interactive stratospheric aerosol models' response to different amounts and altitudes of SO2 injection during the 1991 Pinatubo eruption, Atmos. Chem. Phys., 23, 921–948, https://doi.org/10.5194/acp-23-921-2023, 2023.

Wunderlin, E., Chiodo, G., Sukhodolov, T., Vattioni, S., Visioni, D., & Tilmes, S. (2024). Side effects of sulfur-based geoengineering due to absorptivity of sulfate aerosols. *Geophysical Research Letters*, 51, e2023GL107285. https://doi.org/10.1029/2023GL107285

References:

Baran, A. J. and Foot, J. S.: New application of the operational sounder HIRS in determining a climatology of sulphuric acid aerosol from the Pinatubo eruption, J. Geophys. Res., 99, 25673–25679, https://doi.org/10.1029/94JD02044, 1994.

Fleming, E. L., Newman, P. A., Liang, Q., and Oman, L. D.: Stratospheric temperature and ozone impacts of the Hunga Tonga-Hunga Ha'apai water vapor injection, J. Geophys. Res. Atmos., 129, e2023JD039298, 2024.

Hamill, P., Jensen, E. J., Russeii, P. B., and Bauman, J. J.: The Life Cycle of Stratospheric Aerosol Particles, Bull. Am. Meteorol. Soc., 78, 1395–1410, https://doi.org/10.1175/1520-0477(1997)078<1395:TLCOSA>2.0.CO;2, 1997.

Holton, J. R., Haynes, P. H., McIntyre, M. E., Douglass, A. R., Rood, R. B., and Pfister, L.: Stratosphere-Troposphere Exchange, Rev. Geophys., 33, 403–439, https://doi.org/10.1029/95RG02097, 1995.

Millán, L., Read, W. G., Santee, M. L., Lambert, A., Manney, G. L., Neu, J. L., Pitts, M. C., Werner, F., Livesey, N. J., and Schwartz, M. J.: The Evolution of the Hunga Hydration in a Moistening

Stratosphere, Geophys. Res. Lett., 51, e2024GL110841, https://doi.org/https://doi.org/10.1029/2024GL110841, 2024.

Zhou, X., Dhomse, S. S., Feng, W., Mann, G., Heddell, S., Pumphrey, H., Kerridge, B. J., Latter, B., Siddans, R., Ventress, L., and others: Antarctic vortex dehydration in 2023 as a substantial removal pathway for Hunga Tonga-Hunga Ha'apai water vapor, Geophys. Res. Lett., 51, e2023GL107630, 2024.